# Antenatal care processes in rural Southern Nepal: gaps in and quality of service provision—a cohort study

Emily Bryce ,[1] Joanne Katz,[1] Tsering Pema Lama,[1,2] Subarna K Khatry,[1,2] Steven C LeClerq,[1,2] Melinda Munos[1]

¹Department of International Health, Johns Hopkins University Bloomberg School of Public Health, Baltimore, Maryland, USA
²Nepal Nutrition Intervention Project Sarlahi, Kathmandu, Nepal

**Correspondence to**
Dr Emily Bryce;
ebryce2@jhu.edu

## ABSTRACT

**Objectives** This study aimed to compare a standard quality of care definition to one that reflected focused antenatal care (FANC) guidelines and examine associations with receipt of good quality of care.

**Design** This study was a longitudinal cohort study.

**Settings** Five government health posts in the Sarlahi district of Southern Nepal

**Participants** Pregnant women between the ages of 15 and 49 who presented for their first antenatal care (ANC) visit at the study health posts.

**Main outcomes** There were two quality of care definitions: (1) provision of seven services at least once during pregnancy (QOC1) and (2) provision of services to reflect the FANC guidelines by incorporating a frequency of care dimension for certain services (QOC4+).

**Results** There was variation in service provision both in terms of frequency of provision and by gestational age at the visit. There were 213 women (49.1%) that received good quality care by the first definition, but when the frequency of service provision was included for the second definition the percentage dropped to 6.2%. There were significant differences in provision of quality care by health post for both definitions. The number of visits (QOC1 adjusted risk ratio (aRR) 1.18, 95% CI 1.13 to 1.23; QOC4+ aRR 1.46, 95% CI 1.11 to 2.80) and care during the first trimester (QOC1 aRR 1.22, 95% CI 1.01 to 1.49) and maternal age (QOC1 aRR 1.27, 95% CI 1.03 to 1.58) were associated with greater likelihood of good quality ANC.

**Conclusion** This analysis demonstrated that measuring quality of care by receipt of services at least once during pregnancy may overestimate the true coverage of quality of ANC. Future efforts should improve feasibility of including frequency of care in quality of care definitions.

## Strengths and limitations of this study

► Services received during antenatal care was established using direct observation rather than maternal report, thus eliminating the possibility the risk of recall bias.

► Content of care was observed across the entire pregnancy, rather than cross-sectionally at a single visit.

► The study observers were required to meet inter and intraobserver reliability standards before the study began.

► Potential for observer effect, whereby providers altered their care because of the presence of the study observers.

► Smaller sample size because of the resources required for the longitudinal nature of the study.

Along the reproductive health continuum, antenatal care (ANC) is defined as care provided during pregnancy by a skilled health provider, often at a first level facility.[3] ANC provides the opportunity to identify pregnancy risk factors through screening processes, prevent and manage diseases (pre-existing or pregnancy-related), and to provide health education and promotion.[3] Furthermore, good quality ANC has been associated with higher rates of facility delivery, which improves birth outcomes for mother and infant.[6 7]

Guidelines for ANC have evolved over time, moving from a European model involving an average of twelve visits to the WHO four-visit, focused antenatal care model (FANC) in 2002 to the most current 2016 WHO guidelines requiring a minimum of eight visits during pregnancy.[3 8] However, for many countries the transition to the eight-visit model has not yet been made. For example, Nepal's national protocol still follows the FANC model, which recommends four visits by 4, 6, 8 and 9 months gestational age.[9 10] The country's safe motherhood programme, 'Aama Suraksha', provides financial rewards to women who

## BACKGROUND

Despite the progress made in the previous two decades, in 2017 approximately 810 women died every day from preventable pregnancy and childbirth related causes.[1] The vast majority of these deaths occurred in low and lower-middle income countries. However, there are existing interventions that reduce maternal and neonatal mortality and morbidity that can be delivered during pregnancy.[2–5]

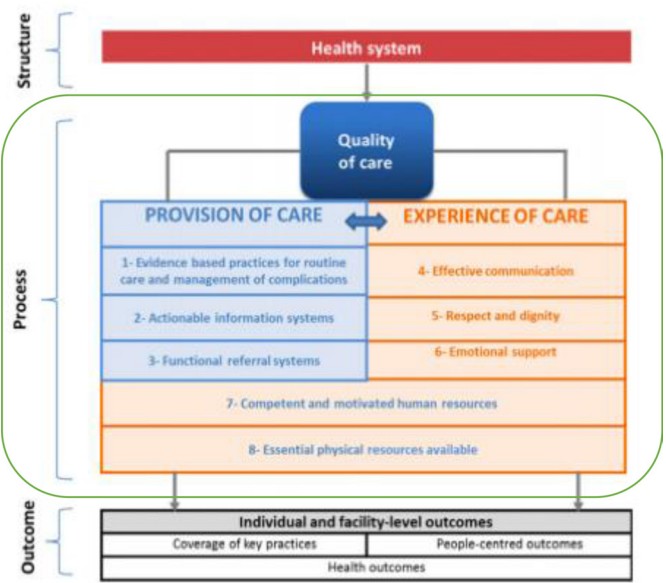

**Figure 1** WHO Quality of Care Framework for maternal and newborn health. Source: Tuncalp *et al*.[14]

attended ANC during those 4 specified months and who deliver in a health facility.[9] Beyond the scheduling recommendations there are suggested components of ANC in Nepal, but there are no current, published guidelines; the last maternity guidelines were published in 1996.[11]

The coverage indicator ANC4+, or the proportion of pregnant women who attend four or more ANC visits, served as the measure of adequate ANC for the fifth Millennium Development Goal and for countries to monitor their maternal care programmes.[12] However, this indicator does not capture the content of the care provided, but rather contact with the health system. This is particularly problematic for ANC, as there are several interventions and health messages required for complete ANC.[3 13 14] Over the years, the measurement of ANC quality has shifted from contact-focused to content-focused.[12]

There is no single definition for 'quality of care' for ANC, which complicates its measurement.[15] The WHO defines quality of care as healthcare that is 'safe, effective, timely, efficient, equitable, and people-centered'.[14] Accompanying this definition is a framework (figure 1), which combines the Donabedian systems approach of evaluation (structure, process, outcome) and the two components of process quality (provision and experience of care) that are central components to other authors' definitions of quality of care.[14 16–19]

In Nepal, 84% of women received any ANC from a skilled provider and 69% of women had at least four ANC visits, though the proportion is lower in rural areas.[20] 58.6% of women had a visit during the 4 months (fourth, sixth, eighth and ninth) per national protocol. Although coverage of ANC has increased, as of 2016 the maternal mortality ratio was equal to 259 deaths per 100 000 live births, well above the Sustainable Development Goal target of 70 deaths per 100 000 live births.[20] This indicates

a gap in quality for provision of maternal care, including ANC.

A few other studies have examined the quality of ANC in Nepal and have shown significant gaps in quality of care, though some of the analyses use Demographic Health Survey data that is cross-sectional and subject to recall bias.[5 21–23] To our knowledge, this is the first study in Nepal to use data generated by a direct observation during a longitudinal cohort study to examine ANC. Furthermore, the longitudinal nature of this study allows us to assess the frequency of service provision, as there are certain services (ie, blood pressure and weight measurement) that should occur at every ANC visit. This study aims to describe the provision of ANC in our study population and to investigate factors associated with high-quality ANC.

## METHODS

### Data sources

The data for this analysis were collected through the Coverage Validation Study, which was conducted within a part of the study area of the Nepal Nutrition Intervention Project Sarlahi located in the Sarlahi District of rural, Southern Nepal. This study is one in a series of validation studies focused on measurement of nutrition-related indicators during pregnancy. The parent study aimed to enrol 450 women to end up with 300 women with complete ANC observation and completed follow-up interview for adequate power for the validation analyses. A total of 434 women were enrolled in the parent study, which represents the sample for this analysis.

Convenience sampling was used to enrol all eligible women as they presented at ANC at the five government health posts. The health posts were selected based on accessibility for the study team and client case load (15–25 patients per week). Pregnant, married women, ages 15 and older who were presenting for their first ANC visit at one of five government health posts selected for this study were eligible to participate. If a woman presented after already attending an ANC visit or ultrasound appointment, she was deemed ineligible because the study would not be able to observe all ANC visits.

A demographic questionnaire was administered at the enrolment visit. The direct observation of the first and all subsequent ANC visits at the study health posts were conducted by trained study staff, using a checklist of 28 items. The socioeconomic and client satisfaction data were collected during a postpartum interview.

### Process quality of care assessment

The aim of this analysis was to evaluate the quality of the processes of care provided during ANC. The process of care includes two components, provision of care and experience of care (figure 1).

Given that Nepal's national protocol is based on four antenatal visits, the FANC model served as the standard for the provision of care. In addition to the FANC model,

**Table 1** Comparison of services included in previous Quality of Care scores

|  | FANC | Nepal MoH annual report | Nepal DHS analyses | Expert survey | What CVS-2 captured |
|---|---|---|---|---|---|
| Provision of care |  |  |  |  |  |
| IFA | X | X | X | X | X |
| IPTp | X |  |  | (not for Nepal) |  |
| TT vaccine | X | X | X | X |  |
| Deworming |  | X | X | X | X |
| BP measured | X | X (2017 report) | X | X | X |
| Weight measured |  |  | X |  | X |
| Blood sample | X |  | X | X | X |
| Urine sample | X |  | X | X | X |
| Counselling | X | X | X | X | X |
| Nutrition | X | X | X | X | X |
| Breast feeding | X |  |  | X | X |
| Facility delivery | X | X | X |  | X |
| Family planning |  |  | X |  |  |
| PP care |  | X |  |  | X |
| Experience of care |  |  |  |  |  |
| Ability to discuss problems |  |  | X | X | X |
| Respectful care |  |  | X | X | X |
| Satisfied with explanation of information |  |  | X | X |  |

BP, blood pressure; DHS, Demographic and Health Survey; FANC, focused antenatal care; IFA, Iron Folic-Acid; MoH, Ministry of Health; PP, Post-partum.

Nepal's Ministry of Health (MoH) annual report, a country-led quality analysis of the 2016 Demographic and Health Survey (DHS) and an expert survey were reviewed to determine final service inclusion (table 1).[22 24–26] The MoH report and 2016 DHS were referenced to include the country's priorities and for comparison to other quality of care analyses. Additionally, the MoH annual report is the most recent documentation available with recommended ANC services in Nepal.

The provision of care domain was measured by receipt of the following: blood pressure measurement, weight measurement, iron folic-acid (IFA) supplementation, blood test prescription, urine test prescription, deworming medication and counselling messages related to nutrition, breast feeding, facility delivery and postpartum care. The prescription of blood and urine tests, rather than the performance of the test, was recorded because only one of the five health posts had a laboratory on-site for use. The parent study focused on nutrition-specific interventions, therefore we did not capture 'counselling on complications and/or warning signs', which is commonly included in antenatal quality of care definitions.[23 27–29] Similarly, how the tetanus toxoid vaccine is distributed in this setting did not allow for its capture during observations and its inclusion in the quality of care metric.

We developed two quality of care definitions (table 2). The process quality for the total study population is defined to align with other analyses of DHS or similarly collected cross-sectional data; at any point during a woman's pregnancy, she received any IFA (non-zero number of tablets), counselling in two of four areas (hypothesised to be dependent on gestational age), and the other five provision of care services at least once (QOC1). For the women who attended four or more ANC visits, quality was measured as at least 120 IFA tablets received, weight and BP measurements at least four times (as these services should be received at each ANC visit), counselling in all four areas and the remaining services at least once (QOC4+). Experience of care was measured the same in both groups.

### Covariates

There were six maternal characteristics that were included to examine their association with quality of care: maternal age, maternal education, prior live births, trimester at enrolment household socioeconomic status (SES) and health post attendance. Maternal education was dichotomised in 'zero years of education' and 'any years of education' and prior live births into 'no prior live births' and 'one or more prior live birth'. The trimester at enrolment was calculated by subtracting the date of reported last menstrual period (LMP) from the date of enrolment. Gestational age at each visit was calculated by subtracting the date of LMP from the visit date. These were then

**Table 2** Quality of care definitions

| | Definition 1: entire study population (QOC1) | Definition 2: women who had recommended 4+ visits (QOC4+) |
|---|---|---|
| Provision of care | | |
| IFA | Any | 120+ (30/visit)* |
| Deworming | Once | Once |
| BP measured | Once | Four times* |
| Weight measured | Once | Four times* |
| Blood sample prescription | Once | Once |
| Urine sample prescription | Once | Once |
| Counselling† | Two of four areas | All four areas |
| Experience of care | | |
| Ability to discuss problems | 'Yes' | 'Yes' |
| Respectful care | 'Yes' | 'Yes' |

*IFA provision, weight and BP measurement are included at each visit in the focused antenatal care model. Other references do not indicate frequency of provision throughout pregnancy.
†Areas of counselling: nutrition, breast feeding, facility delivery and PP care.
BP, blood pressure; IFA, Iron Folic-Acid; PP, Post-partum.

categorised into four groups (<4 months, 4–6 months, 7–8 months and 9+ months) to reflect the FANC model groups. The household SES was constructed through the summation of 11 variables (including number of rooms, fuel and water sources, latrine type and ownership variables) and dividing the total sum by the total number of non-missing responses. This proportion was then separated into quartiles. Nine of the 434 women (2%) attended more than one health post during observations; the remaining 98% of participants attended the same health post for all observations. For those nine women, the health post they attended more frequently was considered their 'primary health post'.

## Statistical analysis

Receipt of services was examined by gestational age using $\chi^2$ tests. Descriptive statistics were estimated to compare quality of care provided at each of the five health posts and between women of different characteristics using proportions for categorical variables and means for continuous variables.

Bivariable and multivariable log-binomial regression models were used to estimate risk ratios and the corresponding 95% CIs for the associations between receipt of (1) QOC1 and (2) QOC4+. Relative risk estimates were calculated instead of ORs because the outcome was not rare (>10% of women received good quality of care) and therefore an OR would have overestimated the

relationships being examined. If the log-binomial regression model did not converge, a poisson model with robust error variance was used. All analyses were conducted using Stata V.14.2 (Stat Corp).

## Patient and public involvement

Study participants were not involved in this design, recruitment, conduct or dissemination of this research. The 28-item checklist was reviewed by a local community advisory board in Nepal before the start of the study, but the public had no other part in the development or implementation of this study. There are no plans to disseminate results to the participants or community, aside from the local study staff who reside in the community.

## RESULTS

A total of 441 women were enrolled in the study and seven women (1.6% of the participants) were lost to follow-up for the 6-month postpartum interview, resulting in a final sample of 434 women. The women were between 16 and 41 years old at enrolment, with an average age of 22.5 years (table 3). Sixty per cent had four or more ANC visits and 31% of women attended a visit during each of the four recommended months outlined by the FANC model. Sixty per cent of the women reported no years of education and 32% had no prior live births.

Frequency of an intervention or counselling message being observed, categorised as never, once or more than once and the reported experience of care is shown in table 4. The majority of IFA and medication-related interventions were received once or more than once. Four women (0.9%) were observed receiving deworming medication twice, which is outside recommendations. The majority of women had blood pressure and weight measured more than once (78.1% and 81.3%, respectively). Over 70% of women did not receive any counselling regarding breast feeding and 67.5% never received advice on postpartum related subjects (eg, importance of postpartum visits, when to come for the visit). Eighty-seven per cent of women reported being able to discuss problems and concerns with their provider and 97.8% reported somewhat or very respectful care.

There were no differences in the measurement of or advice concerning blood pressure or weight by gestational age at the visit (table 5). The proportion of women receiving the service in each of the four categories was significantly different for all other services. A greater proportion of women beyond 4 months gestational age received IFA tablets and were given or told to buy calcium tablets than less than 4 months. Conversely, a higher proportion of women received deworming medication and prescriptions for blood and urine tests at visits that occurred at less than 4 months and between four to 6 months. Nutrition-related advice and counselling on ceasing smoking and drinking was more frequently provided at visits earlier in pregnancy. Advice on matters more related to delivery (breast feeding, delivery in a

| Table 3 | Characteristics of participants | | |
|---|---|---|---|
| Characteristic | Mean | SD | Range |
| Woman's age, years | 22.5 | 4.2 | 16–41 |
| Estimated GA at enrolment, months | 3.5 | 1.3 | 1–9 |
| Total number of visits per woman | 4.5 | 2.5 | 1–14 |
| | **N** | **%** | |
| SES quartiles* | | | |
| 1 | 167 | 38 | |
| 2 | 74 | 17 | |
| 3 | 132 | 30 | |
| 4 | 61 | 14 | |
| Did this woman have any prior live births? | | | |
| Yes | 298 | 69 | |
| No | 136 | 31 | |
| Did the woman have any years of education? | | | |
| No | 259 | 60 | |
| Yes | 175 | 40 | |
| Trimester at enrolment | | | |
| 1–3 months | 190 | 44 | |
| 4–6 months | 236 | 55 | |
| 7–9 months | 7 | 2 | |
| Total number of visits per woman | | | |
| 1 visit | 46 | 11 | |
| 2–4 visits | 205 | 47 | |
| 5–7 visits | 125 | 29 | |
| 8+ visits | 58 | 13 | |
| Number of visits during recommended months (<4, 4–6, 7–8, 9+) | | | |
| 1 | 74 | 17 | |
| 2 | 100 | 23 | |
| 3 | 126 | 29 | |
| 4 | 133 | 31 | |

GA, gestational age; SES, socioeconomic status.

facility) and postpartum subjects were given to a higher proportion of women at 9 months gestational age or greater.

## Quality of care

There were 213 women (49.1%) of the total 434 that were observed receiving good quality of ANC by the first definition (QOC1). The proportion significantly decreased to 10.3% (27 of 262 women) when the second definition was applied among women with four or more visits (QOC4+); 6.2% of the entire cohort. There was significant variation between health posts and the proportion of women receiving good quality of care (figure 2A,B). The second health post, at which the greatest number of women were enrolled, did not have any women with

four or more visits receive good quality of care. The proportion of women that received each element of the quality of care definitions by health post are presented in figure 3A,B to illustrate gaps in provision. Across the majority of the health posts, the element with the lowest prevalence was the receipt of the appropriate number of counselling messages. While any IFA supplementation for QOC1 is not a limiting factor, the proportion of women who received the requisite 120+ tablets for QOC4+ is much lower across all health posts.

Table 6 presents the bivariate and multivariate regression results. The greater the number of ANC visits a woman attended was associated with improved quality of care by both definitions (QOC1 adjusted risk ratio [aRR]=1.18 (1.13 to 2.13, p<0.01); QOC4+ aRR=1.46 (1.11 to 2.80, p<0.01)). Additionally, women who were 20 years old or younger had 27% greater likelihood of receiving QOC1 than older women. Women who attended their first ANC visit during the first trimester were 22% more likely to have received QOC1. The direction and magnitude of these two covariate's associations with QOC4+ were similar to QOC1, but they were not statistically significant.

## DISCUSSION

This paper examined the process and quality of ANC in Southern Nepal using data from a longitudinal study of direct ANC observations. The analyses reveal there are gaps in provision of care and differences in services received by gestational age at visit. Nearly half of women received good quality of ANC defined by receipt of the services at least once during pregnancy and a good experience of care. However, when the QOC definition was expanded to reflect the FANC model recommendations, the percent drops to 6.2% of women receiving good quality of care. While over 60% of the women attended at least four ANC visits, only 31% attended a visit during each of the four recommended gestational age periods as outlined by the FANC model. This illustrates that aside from not capturing the content of care, the ANC4+ indicator may a poor measure of the recommended contacts for measuring ANC coverage.

The results show a gap in care for counselling during ANC, specifically messaging pertaining to breastfeeding and postpartum visits. Low rates of counselling in these areas have been documented in other studies of quality of ANC as well.[27 29 30] The high rate of counselling about nutrition in our study is similar to another in Nepal using DHS data.[23] The lower rates of counselling on facility delivery, breast feeding and postpartum visits in our setting could be because the receipt of financial incentives for the safe motherhood programme is contingent on facility delivery and it is assumed these messages will be delivered at that time. However, rates of facility delivery in Nepal are still quite low, 56% in 2016,[20] therefore these aspects of counselling should also be covered during ANC. Additionally, a recent review also found major gaps in quality of counselling messaging in Nepal and four other South

Table 4 Distribution of provision and experience of care during antenatal care

| Provision of care | Never (%) | Once (%) | More than once (%) |
|---|---|---|---|
| IFA or medicine-related | | | |
| Received IFA tablets | 8.3 | 19.8 | 71.9 |
| Received a deworming | 25.6 | 73.5 | 0.9 |
| Was given or told to buy calcium | 13.8 | 26.3 | 59.9 |
| Physical examination | | | |
| Blood pressure was measured | 2.8 | 19.1 | 78.1 |
| Weight measured | 1.6 | 17.1 | 81.3 |
| Was given a prescription for a blood sample | 6.7 | 59.0 | 34.3 |
| Was given a prescription for a urine sample | 7.6 | 55.5 | 36.9 |
| Counselling | | | |
| Was told why deworming was being given | 93.8 | 6.2 | 0.0 |
| Was told anything about blood pressure measurement | 4.4 | 21.0 | 74.7 |
| Was told anything about weight gain or loss | 2.1 | 19.1 | 78.8 |
| Was told why biospecimen (urine or blood) should be given | 8.5 | 42.9 | 48.6 |
| Was given advice about food or nutrition during pregnancy | 5.3 | 19.4 | 75.3 |
| Was told not drink and/or smoke | 49.8 | 27.9 | 22.4 |
| Was given advice about breast feeding | 70.5 | 7.1 | 22.4 |
| Was told to deliver in a facility | 28.3 | 32.3 | 39.4 |
| Was given advice on post partum-related subjects | 67.5 | 11.8 | 20.7 |
| **Experience of care** | **N (%)** | | |
| Reported being able to discuss problems and concerns | 379 (87.3%) | | |
| Reported level of respect of treatment by health provider | | | |
| Very disrespectful | 2 (0.5%) | | |
| Somewhat disrespectful | 2 (0.5%) | | |
| Neither disrespectful nor respectful | 5 (1.2%) | | |
| Somewhat respectful | 135 (31.0%) | | |
| Very respectful | 290 (66.8%) | | |

IFA, Iron Folic-Acid.

Asian countries.[31] We did not measure the quality of the counselling messaging themselves, only their delivery. However, Torlesse *et al*'s findings may mean that the coverage measures in our population may overestimate the proportion of women who received quality counselling that led to information transfer from provider to client.

Aside from weight and blood pressure measurement, which should be completed at every visit, there were significant differences in the proportion of women receiving a service at a given gestational age. This corresponds with findings from studies in Pakistan and Bangladesh.[28 32] The timing is a critical component of care because certain screening tests (ie, blood test for syphilis) and IFA supplementation should be received earlier in pregnancy.[3] Furthermore, this has implications for women who do not attend ANC during all recommended months and may miss components of care.

The definition of quality of ANC is commonly the receipt of certain services at any point during pregnancy. Our data show that using this definition, 49.1% of the women in our population received good quality of ANC. This is higher than another study using 2011 Nepal DHS data,[23] and studies conducted in India, Bangladesh and sub-Saharan Africa.[27 28 33 34] A possible explanation could be that as our data is more recent, health service delivery has improved since the previous studies. Additionally, the other studies primarily rely on maternal recall of services received during ANC, many of which have yet to be validated, which could lead to reporting errors. Finally, our coverage of QOC1 could be higher due to the observer effect (discussed in full in limitations), resulting in the providers at the study clinics providing more complete care than they would if not under observation.

To our knowledge, this is the first study in a low-income and-middle-income setting to examine quality of care

**Table 5** Receipt by gestational age at visit

| Service observed | Less than 4 months (%) | 4–6 months (%) | 7–8 months (%) | 9+ months (%) | $\chi^2$ p value |
|---|---|---|---|---|---|
| **IFA or medicine related** | | | | | |
| Received IFA tablets | 20.90 | 66.40 | 69.10 | 69.00 | p<0.01 |
| Woman was given deworming tablets | 20.60 | 30.50 | 0.90 | 0.00 | p<0.01 |
| Woman was given or told to buy calcium tablets | 19.70 | 51.10 | 53.20 | 45.60 | p<0.01 |
| **Physical examination** | | | | | |
| Woman's blood pressure was measured | 69.60 | 65.80 | 67.50 | 70.80 | 0.335 |
| Woman's weight was measured | 69.00 | 68.40 | 68.20 | 69.90 | 0.958 |
| Woman was given a prescription for a blood sample | 56.40 | 36.50 | 16.60 | 13.40 | p<0.01 |
| Woman was given a prescription for a urine sample | 56.10 | 37.20 | 17.30 | 13.40 | p<0.01 |
| **Counselling** | | | | | |
| Woman was told why deworming tablet was given | 3.00 | 1.60 | 0.50 | 0.60 | 0.015 |
| Woman was told anything about her BP | 66.00 | 61.20 | 62.70 | 63.80 | 0.479 |
| Woman was told anything about weight/gain/loss | 67.50 | 65.80 | 64.10 | 65.70 | 0.805 |
| Was told why biospecimen (urine or blood) should be given | 53.40 | 41.70 | 25.80 | 24.30 | p<0.01 |
| Woman was given advice about food/nutrition during pregnancy | 72.50 | 64.00 | 61.30 | 60.20 | p<0.01 |
| Received counselling on smoking and/or drinking | 29.60 | 22.30 | 13.10 | 16.70 | p<0.01 |
| Woman was given any advice on breast feeding | 14.30 | 12.10 | 16.40 | 28.90 | p<0.01 |
| Woman was told to deliver in a facility | 28.40 | 25.60 | 27.20 | 61.10 | p<0.01 |
| Received counselling on post partum-related subjects | 14.90 | 13.00 | 15.90 | 28.60 | p<0.01 |

IFA, Iron Folic-Acid.

by including frequency of service provision as outlined by the FANC model. A study in Kenya asked women to report whether they had their weight and blood pressure measured 'never, a few times, most of the time and all the time' and found the majority of women to have received the two services most or all of the time.[29] This is comparable to our findings, but this study did not create a final QOC metric to which we can compare our QOC4+results.

Our analysis showed that the proportion of women received good quality of care differed by health post, more so for QOC4+ than for QOC1. A multi-level model with a random intercept for health post and inclusion of health post level covariates would have been ideal for further elucidating reasons for these differences. However, with only five health posts, the number of clusters was too small to conduct these analyses. Exploring associations between provider characteristics and quality of care would be helpful to explain differences in provision, however the parent study did not collect this information. Distance from facility, which has been shown to be associated with ANC attendance, was also a potential influential factor that we were unable to capture.

The total number of ANC visits is the only characteristic that is significantly associated with both of the QOC definitions, which is consistent with another study.[33] The greater the number of visits a woman attended, the greater the opportunity to have received all necessary components. Younger women (<20 years) were more likely to have received better quality of care, which was not the case in other studies in Nepal and Kenya.[23 29]

There were no differences in QOC by SES quartile in our study, whereas others have shown there to be a significant

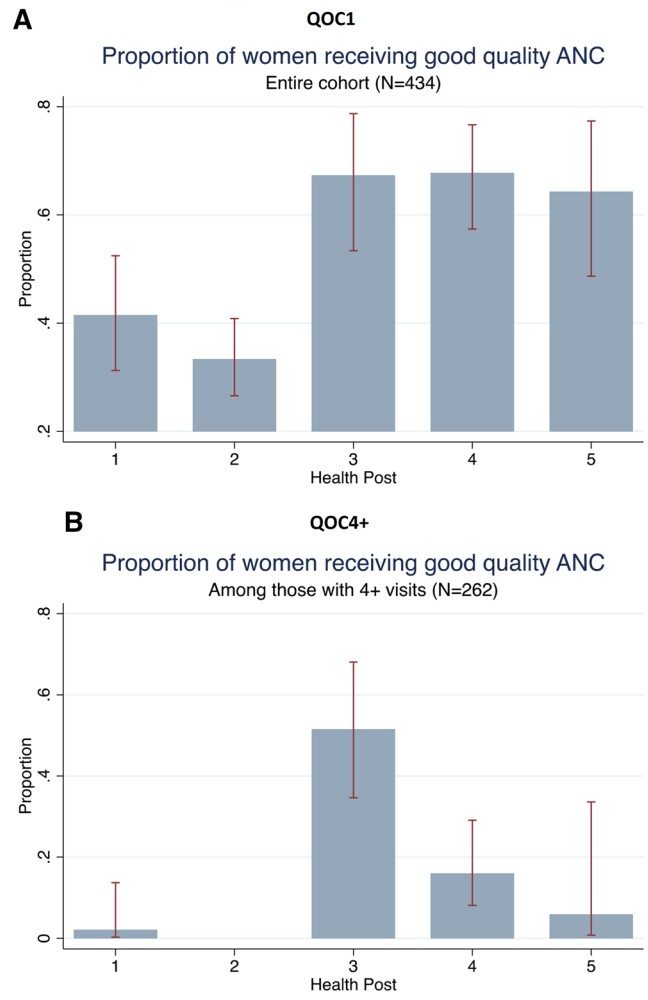

**Figure 2** (A) QOC1 and (B) QOC4+. ANC, antenatal care.

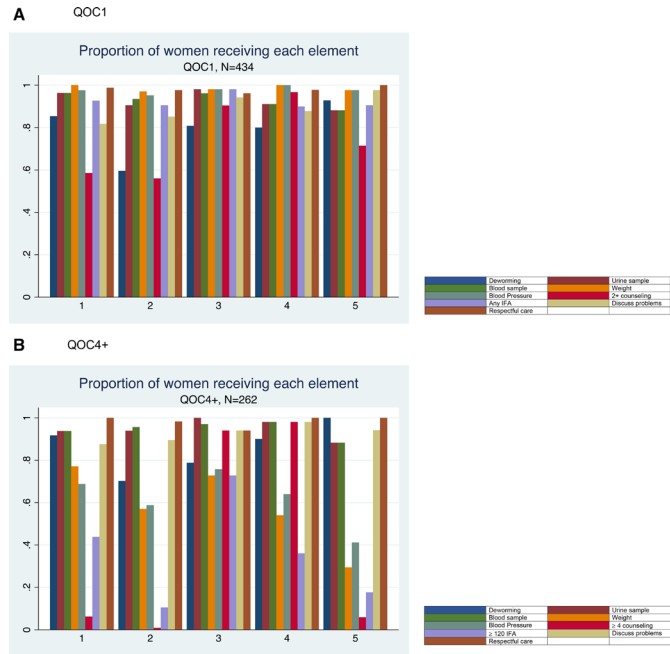

**Figure 3** (A) QOC1 and (B) QOC4+.

relationship between SES and QOC.[23 33 35] However, these studies primarily use data from population-based surveys (eg, DHS) and have a more heterogeneous population in regard to SES. In our population, the mean score was 6.3 (SD=2.3) and was skewed towards lower scores. There was no difference in QOC1 by education, which has been demonstrated in some studies[29 36] but not others.[23 27 33] As with SES, our population was more homogenous in educational attainment; 60% of the population reported no years of education and only one woman reported schooling beyond grade school, compared with 38.9% with no education and 7% with tertiary education in the 2011 Nepal DHS study.[23]

There were no statistically significant associations between covariates and QOC4+, which could be due to a lack of power (discussed further in limitations), as the magnitude of the coefficient for many is quite large. While we cannot draw conclusions on associations between the covariates and the QOC4+ outcome, the descriptive analyses demonstrate the gaps in care and difference in coverage between QOC4+ and QOC1. The health post with the greatest number of patients had no women with high-quality QOC4+, driven primarily by a lack of counselling. This health post also had the smallest proportion of women who received 120+ tablets and deworming medication. The increased patient load could explain the low proportions of women receiving the commodity-based services; stockouts might have occurred, or the providers may have limited the number of IFA tablets given to a woman at any one visit to delay a stockout. The lack of counselling could be due to time limitations; the provider may feel that they cannot spend adequate time to convey all the health education messaging because of waiting patients. However, the satisfaction score for this health post was high and similar to the other posts.

A strength of this study is that the receipt of services was collected by direct observation, rather than maternal report. The use of direct observation removes the possibility for recall bias and social desirability bias from the participant, both of which can be present in the cross-sectional studies commonly used in antenatal quality of care studies. Furthermore, the longitudinal design of the parent study allowed for the observation of a woman's entire pregnancy and record of frequency of service provision. Another strength includes the use of trained observers who were required to meet interobserver and intraobserver reliability standards before the study began.

There are limitations to this analysis as well. While the data is not subject to recall or social desirability bias, the presence of the study observers during ANC could have elicited an observer effect, where the providers knew they were being observed and altered their care. To reduce this possibility, providers were told during the consent process that observers are not medically trained and will not know if the procedures they are observing are correct or not, nor will they report what they observe to the clinic supervisor. Additionally, the observers were present for over a year and one would hope that any observer effect

**Table 6** Regression results

| | Definition 1 (QOC1): total study population (N=434) | | Definition 2 (QOC4+): among women with 4+ visits (N=262) | |
| --- | --- | --- | --- | --- |
| | Bivariate model | Multivariate model | Bivariate model | Multivariate model |
| | RR (95% CI) | aRR (95% CI) | RR (95% CI) | aRR (95% CI) |
| Maternal age ≤20 | 1.14 (0.94 to 1.38) | 1.27* (1.03 to 1.58) | 0.98 (0.47 to 2.02) | 1.53 (0.69 to 3.43) |
| No prior live birth | 1.00 (0.82 to 1.24) | 0.81 (0.64 to 1.02) | 0.70 (0.31 to 1.60) | 0.60 (0.23 to 1.56) |
| Any years of education | 1.10 (0.91 to 1.34) | 0.96 (0.80 to 1.16) | 1.05 (0.51 to 2.16) | 0.62 (0.31 to 1.26) |
| Number of visits | 1.07** (1.04 to 1.11) | 1.18** (1.13 to 1.23) | 0.96 (0.80 to 1.15) | 1.46** (1.11 to 2.80) |
| Enrolment during first trimester | 1.03 (0.85 to 1.25) | 1.22* (1.01 to 1.49) | 1.07 (0.52 to 2.18) | 1.15 (0.63 to 2.08) |
| SES quartiles (ref: 1st) | | | | |
| 2 | 1.27 (0.97 to 1.67) | 1.16 (0.92 to 1.47) | 0.67 (0.19 to 2.42) | 1.77 (0.52 to 6.03) |
| 3 | 1.21 (0.95 to 1.54) | 0.99 (0.79 to 1.24) | 1.77 (0.76 to 4.10) | 1.48 (0.64 to 3.40) |
| 4 | 1.31 (0.99 to 1.74) | 1.09 (0.82 to 1.45) | 1.22 (0.39 to 3.82) | 1.17 (0.37 to 3.71) |
| Health post (ref: 1) | | | | |
| 2 | 0.80 (0.56 to 1.12) | 0.66* (0.48 to 0.91) | −† | −† |
| 3 | 1.62** (1.18 to 2.23) | 1.69** (1.23 to 2.30) | 24.73** (3.46 to 176.86) | 36.11** (5.03 to 259.10) |
| 4 | 1.63** (1.22 to 2.19) | 1.84** (1.40 to 2.41) | 7.68* (1.00 to 59.11) | 12.59* (1.67 to 94.82) |
| 5 | 1.55* (1.10 to 2.18) | 1.95** (1.39 to 2.73) | 2.82 (0.19 to 42.70) | 6.22 (0.47 to 81.92) |

*p<0.05; **p<0.01.
†There were no women who received good quality of care by the QOC4+ definition.
aRR, Adjusted Risk Ratio; RR, Risk Ratio; SES, socioeconomic status.

would dissipate over the extended time period. The client satisfaction data was collected at the postpartum interview, which means it is subject to recall and social desirability bias. However, there has been evidence of 'courtesy bias', where individuals are less likely to report critically at the facility, which is avoided in this case. Additionally, gestational age estimation was based on maternal report of LMP, which is subject to recall bias as well. Eighty-one per cent of the sample presented by the fourth month of pregnancy, so the hope is that with a shorter recall period the LMP data is still fairly accurate. Another limitation was that we only observed women who presented for ANC at government health facilities. Therefore, the generalisability may be limited to women who attend ANC or those who remain within the system in which we observed (excluding private facilities, traditional healers, etc). In 2016, 16% of women did not receive ANC for a skilled provider, indicating that there is a substantial proportion of the population to which these findings may not be generalisable.[20] A final limitation is the smaller sample size compared other ANC QOC studies, which may have led to limited power in detecting associations between QOC and maternal and facility characteristics. However, because the longitudinal direct observation design is so time and resource intensive, it would be difficult to get a much larger sample size.

This analysis demonstrates that measuring quality of care by receipt of services at least once during pregnancy may be overestimating the true coverage of quality of ANC. Furthermore, as countries transition to the eight-visit 2016 WHO ANC recommendations, the degree of underestimation will only increase. Additionally, despite a common belief that commodity-based services such as IFA tablets are often the limiting factor for quality of care, in this study the largest gap in service provision was seen for adequate counselling messaging and the counselling adequacy varied significantly across the study health posts. Future programming and policy efforts should fortify provider training on effective counselling delivery and interpersonal skills to close the gap in counselling provision. This analysis supports the continued research and programmatic efforts in strengthening of routine health information systems and electronic health records to measure quality of care over a woman's entire pregnancy. This would allow the frequency of service provision to be included in quality of care assessment. Additionally, these systems and household surveys should be updated to capture additional indicators of counselling delivery. The inclusion of frequency of specific services provided in ANC in quality of care measurement may be the next step in closing the 'quality-coverage' gap and achieving optimal maternal, newborn and child survival.

**Contributors** MM and JK designed the study. TP, SKK, SCL, JK and EB contributed to the implementation of the study. EB conducted analyses and wrote the first draft of manuscript. MM and JK submitted edits to the manuscript. MM, JK, TP, SKK, EB and SCL all have read and approve the final version of the manuscript. JK acted as guarantor

**Funding**  This research was funded by the Bill and Melinda Gates Foundation (grant number OPP1172551).

**Competing interests**  None declared.

**Patient and public involvement**  Patients and/or the public were involved in the design, or conduct, or reporting, or dissemination plans of this research. Refer to the Methods section for further details.

**Patient consent for publication**  Not applicable.

**Ethics approval**  This study involves human participants and was approved by The Institutional Review Board of the Johns Hopkins Bloomberg School of Public Health (Baltimore, Maryland, USA) and the National Health Research Council (Kathmandu, Nepal) approved this study. Informed consent was obtained from providers and participants at the beginning of the study and participants were re-consented at the postpartum interview. Participants gave informed consent to participate in the study before taking part.

**Provenance and peer review**  Not commissioned; externally peer reviewed.

**Data availability statement**  Data are available upon reasonable request.

**ORCID iD**
Emily Bryce http://orcid.org/0000-0002-4823-1647

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
