## [Reviewer comments · BMJ Open]

ARTICLE DETAILS

TITLE (PROVISIONAL)	Antenatal care processes in rural Southern Nepal: gaps in and quality of service provision: a cohort study
AUTHORS	Bryce, Emily; Katz, Joanne; Plama, Tsering; Khatry, Subarna; LeClerq, Steven; Munos, Melinda

VERSION 1 – REVIEW

REVIEWER	Rinaldo Zanini Alessandro Manzoni Hospital, mother and child health
REVIEW RETURNED	06-Sep-2021

GENERAL COMMENTS	“Antenatal care processes in rural Southern Nepal: gaps in and quality of service provision” ++ Good and original paper about ante natal care in Nepal. The direct – on the field – observation make, to me, this work very interesting. This kind of research is very usefull towards a reduction maternal mortality in the world. +I appreciate the use of 2 definitions of quality of care. The differences between the two are important to understand the quality of the care in the real world. ++ I think that a brief description – with data on maternal mortality - of Nepal Health services could help a better understanding of the enviromental background of the paper. + What average proportion of pregnant women use the health posts compared with no care (traditional healers) or private services? ++ I think useful to explain how was made the women selection of the 434 women and of the selection of the five government health posts. +Are the choosen health posts in urban/village or agricultural area? +What the proportion of attending pregnant women at 5 health post with the whole population living in the area? + How many government health posts in South Nepal? + How many women average each health post controll in a year? + There are some campaign to recruit the pregnant women. If yes any differences among the 5 government health post? ++ Could be possible to know, for each government health post, the average distance with the women’s residences. Distances and transport facility could play a role in the differences of attending women among the health posts. + Time spent to arrive from home to the health post could be a covariate ++ It’s important to elaborate on the principal reasons for the differences among the 5 health posts for understanding why and to obtain some elements for improvement. I think this issue is important because – due to the low education level of women – the quality and completeness of services offered is important.
--

REVIEWER	Setegn Fenta Debre Tabor University, Statistics
REVIEW RETURNED	04-Oct-2021

GENERAL COMMENTS	Comments and questions for authors  1. English language needed and the authors must follow BMJ journal reference style. 2. The authors add some background about antenatal care processes, quality of service provision and gap from the world to the study area. 3. The authors drop some important variable associated with quality of care. For example distance to health facility, presence of transportation, education level of husbands, occupation of husbands and residence. 4. Statistical analysis must be clearly identified including the formula. 5. Authors must be adding which types of statistical software used. 6. Is there the authors test over dispersion and excess zeros? 7. Discussion part in the first paragraph the authors does not discuss with another's paper.
--

VERSION 1 – AUTHOR RESPONSE

Reviewer: 1

Dr. Rinaldo Zanini, Alessandro Manzoni Hospital

Comments to the Author:

“Antenatal care processes in rural Southern Nepal: gaps in and quality of service provision”

++ Good and original paper about ante natal care in Nepal. The direct – on the field – observation make, to me, this work very interesting. This kind of research is wery usefull towards a reduction maternal mortality in the world.

Thank you

++I appreciate the use of 2 definitions of quality of care. The differences between the two are important to understand the quality of the care in the real world.

Thank you

++ I think that a brief description – with data on maternal mortality - of Nepal Health services could help a better understanding of the enviromental background of the paper.

MMR in the 6th paragraph of the introduction

+ What average proportion of pregnant women use the health posts compared with no care (traditional healers) or private services?

In 6th paragraph of introduction, states that 84% receive ANC from skilled health provider, but the DHS does not disaggregate by public versus private. Due to the Safe motherhood program outlined in paragraph 3 of introduction, anecdotally the majority go to public health posts for ANC and will attend a private facility for an ultrasound, but less so for routine care.

++ I think useful to explain how was made the women selection of the 434 women and of the selection of the five government health posts.

Updated methods section with this information (convenience sampling at the health posts)

+ Are the chosen health posts in urban/village or agricultural area?

Added clarification in the methods (in a very rural, agricultural area)

+ What the proportion of attending pregnant women at 5 health post with the whole population living in the area?

A very small portion. Last population measurement for Sarlahi was in 2011 estimated a population of ~770,000 (compared to our 434 women). The authors don't think this is necessary to include this in the manuscript, as it's not informative to our study.

+ How many government health posts in South Nepal?

As of 2020, 94 health posts in Sarlahi district. The authors did not include this because HP were purposively selected, therefore estimating a probability of selection isn't relevant

+ How many women average each health post control in a year?

Added information on case loads (though, per week not per year as this is the information we collected from the health posts during design to be able to estimate how long enrollment would take).

+ There are some campaign to recruit the pregnant women. If yes any differences among the 5 government health post?

There was no campaign to recruit pregnant women. Women were enrolled in the study when they presented for their first ANC visit.

++ Could be possible to know, for each government health post, the average distance with the women's residences. Distances and transport facility could play a role in the differences of attending women among the health posts.

Unfortunately, we did not collect this information. The parent study (validation study) that this data was collected during did not need this variable and thus we do not have it. The authors agree that this would have been an interesting factor, though more so related to the likelihood of ANC attendance and number of visits rather than the content of the visits themselves.

+ Time spent to arrive from home to the health post could be a covariate

See above response

++ It's important to elaborate on the principal reasons for the differences among the 5 health posts for understanding why and to obtain some elements for improvement. I think this issue is important because – due to the low education level of women – the quality and completeness of services offered is important.

Agreed- in the discussion we lamented on the fact that due to sample size restrictions, we were unable to utilize a multi-level model to examine facility-level factors.

Reviewer: 2

Mr. Setegn Fenta, Debre Tabor University

Comments to the Author:

Comments and questions for authors

1. English language needed and the authors must follow BMJ journal reference style.

Unclear- language is English and the reference style is the modified Vancouver with 3 authors listed and et al if there are 4 or more for the remaining.

2. The authors add some background about antenatal care processes, quality of service provision and gap from the world to the study area.

Added a brief sentence in introduction about background on antenatal care quality in Nepal, but the authors believe that the methods explanation of how the definitions were developed gives a

good background into antenatal care processes (under header “process quality of care assessment) and do not want to repeat themselves in the introduction

3. The authors drop some important variable associated with quality of care. For example distance to health facility, presence of transportation, education level of husbands, occupation of husbands and residence.

The parent validation study did not collect this information as they were not covariates of interest, so unfortunately they could not be included in the model.

4. Statistical analysis must be clearly identified including the formula.

The statistical analysis is clearly stated in the methods; the formula is not a requirement of the journal

5. Authors must be adding which types of statistical software used.

Clearly states Stata 14.2 was used in methods

6. Is there the authors test over dispersion and excess zeros?

The authors did not... It was our impression that this check was for count data, and even though we used poisson data to estimate relative risk when the log-binomial model would not converge, the outcome was binary not count.

7. Discussion part in the first paragraph the authors does not discuss with another’s paper.

For the first discussion paragraph the authors like to give a summary of the manuscript results. The following paragraphs compare these findings to others published.

VERSION 2 – REVIEW

REVIEWER	Rinaldo Zanini Alessandro Manzoni Hospital, mother and child health
REVIEW RETURNED	16-Nov-2021

GENERAL COMMENTS	Thank you for your consideration of some points of my review. I think, anyway, that, at least, you ought touch – few words - these 2 issues: 1) The proportion of women attending to ANC services and 2) “ The authors agree that this would have been an interesting factor, though more so related to the likelihood of ANC attendance and number of visits rather than the content of the visits themselves. ...” I consider interesting to touch on also if you do not have any information or any data about the matter when you discuss limitation. In both the cases I think that, in this way you help the reader to consider your work as a tool to improve the service of ANC and, eventually, to design a new specific research
--

VERSION 2 – AUTHOR RESPONSE

Thank you for your consideration of some points of my review. I think, anyway, that, at least, you ought touch – few words - these 2 issues:

1) The proportion of women attending to ANC services and 2) “ The authors agree that this would have been an interesting factor, though more so related to the likelihood of ANC attendance and number of visits rather than the content of the visits themselves. ...”

I consider interesting to touch on also if you do not have any information or any data about the matter when you discuss limitation.

In both the cases I think that, in this way you help the reader to consider your work as a tool to improve the service of ANC and, eventually, to design a new specific research

Added the DHS 2016 statistic for % of women who never received ANC from a skilled provider. Included a sentence on inability to collect distance data in discussion as well.